# Testicular Germ Cell Tumours and Proprotein Convertases

**DOI:** 10.3390/cancers14071633

**Published:** 2022-03-23

**Authors:** Aitziber Velado-Eguskiza, Laura Gomez-Santos, Iker Badiola, Francisco José Sáez, Edurne Alonso

**Affiliations:** 1Department of Cell Biology and Histology, Faculty of Medicine and Nursing, University of the Basque Country (UPV/EHU), 48940 Leioa, Spain; avelado004@ikasle.ehu.eus (A.V.-E.); laura.gomez@ehu.eus (L.G.-S.); iker.badiola@ehu.eus (I.B.); francisco.saez@ehu.eus (F.J.S.); 2Nanokide Therapeutics SL, 48940 Leioa, Spain; 3Department of Cell Biology and Histology, Faculty of Pharmacy, University of the Basque Country (UPV/EHU), 01006 Vitoria-Gasteiz, Spain

**Keywords:** testicular cancer, Testicular Germ Cell Tumours (TGCT), seminoma, spermatogenesis, Proprotein Convertases (PCs)

## Abstract

**Simple Summary:**

Despite the high survival rate of the most common neoplasia in young Caucasian men: Testicular Germ Cell Tumors (TGCT), the quality of life of these patients is impaired by the multiple long-term side effects of their treatment. The study of molecules that can serve both as diagnostic biomarkers for tumor development and as therapeutic targets seems necessary. Proprotein convertases (PC) are a group of proteases responsible for the maturation of inactive proproteins with very diverse functions, whose alterations in expression have been associated with various diseases, such as other types of cancer and inflammation. The study of the immune tumor microenvironment and the substrates of PCs could contribute to the development of new and necessary immunotherapies to treat this pathology.

**Abstract:**

Testicular Germ Cell Tumours (TGCT) are widely considered a “curable cancer” due to their exceptionally high survival rate, even if it is reduced by many years after the diagnosis due to metastases and relapses. The most common therapeutic approach to TGCTs has not changed in the last 50 years despite its multiple long-term side effects, and because it is the most common malignancy in young Caucasian men, much research is needed to better the quality of life of the many survivors. Proprotein Convertases (PC) are nine serine proteases responsible for the maturation of inactive proproteins with many diverse functions. Alterations in their expression have been associated with various diseases, including cancer and inflammation. Many of their substrates are adhesion molecules, metalloproteases and proinflammatory molecules, all of which are involved in tumour development. Inhibition of certain convertases has also been shown to slow tumour formation, demonstrating their involvement in this process. Considering the very established link between PCs and inflammation-related malignancies and the recent studies carried out into the immune microenvironment of TGCTs, the study of the involvement of PCs in testicular cancer may open up avenues for being both a biomarker for diagnosis and a therapeutic target.

## 1. Introduction

Testicular Germ Cell Tumours (TGCTs) have been of historic relevance for various reasons. Ninety percent of patients diagnosed with metastatic TGCT in 1946 died within a year. Now, however, that statistic has been reversed and the vast majority of patients are cured [1,2]. As a result, TGCT has been proposed as a potential model to find a therapy for the rest of the cancer types [3]. In fact, TGCT tumours played an invaluable role in the establishment of cancer as a stem cell disease. Perhaps because of the aforementioned survival rate, no one has developed better drugs for the treatment of these disorders since the 1970s, even though the most used therapeutic strategies have harmful long term consequences [4,5].

Regardless, there is currently an active push for more research regarding TGCTs to minimize the harmful effect this disease family has on the lives of young men all over the world and that is reflected with the publication of several review papers around that general topic [6,7]. In this review, we also aim to explore the current status of TGCTs in a molecular and clinical setting. We propose to approach the proprotein convertase (PC) family as possible molecules of interest in this disease, since both subjects may be linked to a proinflammatory tumour microenvironment (TME).

## 2. Testicular Germ-Cell Tumours (TGCT)

Germ Cell Tumours (GCTs) originate from germ cells, usually in the gonads (testes and ovaries), but they can also be found elsewhere in the central nervous system, pelvis, thorax, abdomen and mediastinum [8]. TGCTs are the second most common form of Germ Cell Tumour after benign ovarian teratomas and account for more than 90% of neoplasms found in testicles [8,9]. Their incidence has been on the rise since the 1970s, and even though they have an extremely high survival rate, it shrinks considerably by 30 years after the diagnosis, due to metastases and relapses (15–30% of patients) [10,11,12,13]. Taking that into consideration, it is important to study these tumours and understand their development as well as the best clinical paths for their treatment.

The different types of TGCTs have usually been classified following morphological criteria into two categories: seminomas and non-seminomatous Germ Cell Tumours (NSGCTs). Of those two, NSGCTs are less common but more aggressive and heterogeneous. They are categorised into four histological types: embryonal carcinoma, yolk-sac carcinoma, choriocarcinoma and teratoma; most tumours present more than one cell category, making them harder to treat. 

Analysing TGCT incidence by age, two different spikes can be identified (Figure 1). The first peak is in patients aged 25–29 years old, even though these demographical groups have a higher incidence of other malignancies as well. That first peak is due to non-seminomas, mainly yolk-sac tumours and teratomas. The second peak is due to seminomas and is observed between 30 and 39 years of age. Furthermore, there is a clear difference of almost 10 years between the peak incidence of non-seminomas and seminomas [8].

### 2.1. Tumour Classification

An early problem in the study of TGCTs was the different terminology used for their classification by different research groups. Until recently, their classification was morphology-based but it was updated in 2016 by the World Health Organisation (WHO) to reflect the latest studies and the epidemiological similarities or differences between certain types of tumours [14] (Table 1). According to this nomenclature, Testicular Germ Cell Tumours are classified based on their relationship with the germ cell neoplasia In Situ (GCNIS): a group of malignant intratubular germ cells with seminoma-like morphology that appear in the spermatogenic niche and precede most TGCTs. Then, GCTs are classified in two main groups: GCNIS-derived tumours and not GCNIS-derived tumours [14]. 

The most closely related to GCNIS would be pure seminomas, which maintain the same histology. From there, another type of classification that is often used in clinics arises, which divides tumours into seminomatous or non-seminomatous. This is especially important for treatments because seminoma cells are sensitive to chemotherapy and radiotherapy [15,16,17]. 

The distinction between morphology and epidemiology-based classifications is most notably seen in the case of teratomas and yolk sac tumours. There is no significant morphological difference between the pre- and post-pubertal types, but the tumours seen in children more often lack the markers associated with GCNIS that are prevalent in tumours seen after puberty. Therefore, although they were previously considered to be a single disorder, we now know that there are considerable epidemiological and molecular differences [14]. 

In fact, the most prominent theory on the origin of GCNIS states that there is some kind of tumorigenic event in utero during embryogenesis, which impairs the maturation of primordial stem cells into spermatogonia [18]. Those cells are identified as GCNIS and remain unchanged until puberty, when alterations in their molecular environment lead to malignancy. GCNIS could be considered as the precursor of most post-pubertal TGCTs and is usually treated as a stage 0 cancer. 

The seminomatous/non-seminomatous nomenclature is still used today and normally differentiates between pure seminoma (seminomatous tumours) and every other type of TGCT (non-seminomatous) [19]. For the treatment of mixed tumours the non-seminomatous approach has a better prognosis, as these types of cancer present more aggressive characteristics [19]. 

#### 2.1.1. Seminoma

Seminoma is the most common TGCT, accounting for 50% of all cases beyond puberty [9]. In fact, this malignancy is fairly uncommon during infancy and in people older than 50 years, even though it continues to be the most common form of TGCT in the latter, as the incidence of the rest of TGCTs decreases further [9,20]. 

As is the case with all single cell-type tumours, all pure seminoma cells have the same reaction to the therapeutic strategies used on them, with radiotherapy being the most effective. In addition, about 80% of diagnoses are made during stage I, so the survival rate of seminoma patients is exceptionally high [12,21]. 

However, it is important not to underestimate the severity of seminoma, especially with regard to the long-term effects of the treatments used, both on the physical and mental health of patients. The main challenge for seminoma patients is not to improve survival per se, but to develop sensitive and specific biomarkers to safely identify this less threatening form of cancer and thus avoid the overtreatment of patients, mainly with cisplatin-based chemotherapy [22]. Given that seminoma patients are often young men, the focus of research should be on preserving their quality of life and reproductive health. 

#### 2.1.2. NSGCTs

Non-seminomatous Germ Cell Tumours, as mentioned before, are classified into four main histological categories: embryonal carcinoma, yolk-sac tumours, choriocarcinoma, and teratoma. Nevertheless, these tumours do not usually appear in a pure form, and most NSGCTs are mixed cell tumours [23]. As a consequence, the histological analysis of the primary tumour is of great importance to determine which cell types are present and in what proportion. That information is important for the diagnosis, prognosis and treatment selection. 

Similar to seminoma, 50% of NSGCTs are initially diagnosed as stage I cancer [24]. In addition, if left untreated, around 30% of patients will relapse due to previously unnoticed metastases to the retroperitoneal lymph nodes [25]. 

### 2.2. Risk Factors

Gonad developmental processes and genetic predisposition have been described as key factors associated with TGCT, which explains why they are so common amongst younger people.

The main risk factor identified, especially in the case of seminoma, is cryptorchidism (problem with the descent of the testes, leaving one or both of them outside the scrotum). The prevalence of TGCT increases between 3.7- and 7.5-fold in these patients; in fact, between 5% and 10% of testicular cancer patients have a history of cryptorchidism [26,27]. This is probably because the testicle is in an unfavourable molecular environment for its development, which can lead to GCNIS [9].

Other prominent risk factors include a previous tumour in the contralateral testicle, especially in the preceding 5 years (24.5- to 27.5-fold increase) with a family history of TGCT, which has been linked to various candidate genes, some of which are also related to cryptorchidism [9,28,29,30]. 

It has been observed that the cells from GCNIS are really hard to distinguish from germ cells in intersex gonads, which present a delay in their maturation. One theory suggests that these abnormal cells become GCNIS, which would explain why between 25% and 30% of people with gonadal dysgenesis and a Y chromosome develop a GCT [14,31]. 

Lastly, an increased risk of developing a TGCT has also been linked to abnormalities of the germ cell lines that lead to hypofertility and infertility [32,33]. Nevertheless, it is not yet clear whether there is a direct causal relationship or a third element that leads to both impaired fertility and an increased risk of TGCT [34].

A considerable part of TGCTs could be due to environmental factors. In fact, it has been proven that exposure to certain environmental factors during foetal development or early infancy leads to the appearance of GCNIS from gonocytes with impaired maturation, as previously described in people with intersex characteristics [35,36]. The relationship between impaired gonadal development and GCNIS-mediated tumorigenesis is evident, as a lot of these tumours share morphological characteristics like deficient spermatogenesis, tubular shrinking, peritubular sclerosis, immature Sertoli cells, interstitial expansion, hyalinised tubules and microlithiasis [37,38]. 

In summary, the data suggest that TGCTs are a developmental type of cancer that adheres to a “gene-environment” model, in which there are obvious genetic and epigenetic components that lead to tumorigenesis under the right (or wrong) environmental conditions [39].

### 2.3. Diagnosis, Prognosis and Treatment

When a suspicious testicular mass is discovered, the first step is to analyse certain tumoral biomarkers and carry out a trans-scrotal ultrasonography. If the imaging shows a mass that is suspected of being malignant, an inguinal orchiectomy would be carried out for further diagnosis. It is usually recommended not to damage the scrotum during the procedure, as it can lead to complications during treatment, but recent meta-analyses discard these negative effects at least for short-term survival [40,41]. 

The most common primary therapeutic approach, a radical orchiectomy, is performed when the affected testicle is completely extracted [19]. That leads to the loss of the entire organ, which is problematic if the contralateral testicle has already been removed or in young patients. Therefore, in certain cases, research groups have proposed partial orchiectomies or testis-sparing surgery (TSS) to minimise endocrine and psychological negative effects and preserve fertility [42]. TSS is proposed especially in cases of suspected benign tumours and in relapses when the contralateral testicle has already been removed to maintain endocrine function. This practice has been regarded as unnecessary or dangerous due to the risk of relapse compared to radical orchiectomy, but a meta-analysis of multiple studies has shown a relapse rate of only 7.5%, which disappears when adjuvant local radiotherapy is applied (except in radio-resistant teratomas) [41]. 

If the initial diagnosis was stage I cancer, the same biomarkers investigated at the start of the diagnosis are followed up until normalisation. If levels do not return to normal, some kind of metastasis is confirmed, and the diagnosis shifts to a later stage. Adjuvant therapies are considered depending on the diagnosis and evolution of the serum biomarker. All of the information is taken together, and patients are classified into three main categories of prognosis: good, intermediate or poor. This classification (Table 2) was developed by the International Germ Cell Cancer Collaborative Group (IGCCCG) in 1997 [43]. 

The most common paths after orchiectomy are the following: active surveillance, systemic therapy and retroperitoneal lymph node dissection (RPLND). Systemic therapeutic strategies like chemotherapy are usually reserved for higher risk cases when there is already metastasis, due to the possibility of further complications for the patients [5]. RPLND is preferred as the treatment for stage I or II teratomas but is also avoided when possible because of its frequent damaging effects on the reproductive health of the patients [44]. Multiple studies have been carried out to assess the risks associated with the different therapeutic approaches, and a recent meta-analysis published in 2021 by Pierorazio et al. [45] summarises these studies and underlines the importance of considering the long-term toxicities when choosing the best path forward from case to case. 

### 2.4. TGCT Molecular Markers

Molecular markers are useful to track the progress of the treatments used and distinguish between different types of tumours during the diagnostic process. These can be detected through histological analysis by biopsy and by biochemical tests of blood or urine samples. In the case of TGCT, decisions are made based on the patient’s family history and the presence of a few markers with limited diagnostic and prognostic value which have been the same for decades [46]. Further study of this field could lead to a finer sorting of patients according to the potential risks of each treatment. In addition, more precise and less aggressive treatments than cisplatin-based chemotherapy could be developed because this is the treatment that is widely used today, whose risks and side effects are really negative for the patient [5,22].

#### 2.4.1. Histological Markers

One of the easiest forms of tumour analysis is the study of biopsies. In this way, a sort of snapshot is taken of a section of the tumour at a specific point in time. These sections can be analysed using immunohistochemical tools to look for specific molecules. One of the most searched molecules is the transcription factor OCT¾ (also known as POU5F1 and Oct4), which is regarded as one of the key regulators of pluripotency and is specific to primordial germ cells [47,48]. If detected in adult testes, it implies the presence of GCNIS, a seminoma or embryonal carcinoma [47,48]. 

Ninety per cent of TGCTs across all different types have been observed to present an isochromosome of 12p or i(12p) [49]. Even in the other 10% of cases, there is usually an excess of genetic material in the p arm of chromosome 12, which implies a relationship between the genes encoded there and TGCT, both of gonadal and extragonadal origin [50]. Furthermore, this chromosomal anomaly is also present in in situ carcinomas like GCNIS, so it seems to be useful as a biomarker during the first stages of tumorigenesis or even its cause [51]. In addition, a higher number of copies of i(12p) is connected to more aggressive tumours, so this marker can provide information about both diagnosis and prognosis [52]. Some genes present in i(12p) that have been observed to appear mutated in different GCT and linked to TGCT development are the oncogene KRAS and the protooncogenes KIT and NRA [53]. 

Some problems make the use of histological markers in cancer more difficult. First, there is a need for a biopsy, which implies an invasive procedure and at least certain short-term risks. In addition, biopsies only provide information from a certain section of the tumour which, due to tumour heterogeneity, is not always representative of the whole, and from a precise moment in time, which limits their prognostic value. Therefore, the limited biomarker search for TGCT has recently focused on molecules present in blood and plasma, which is considered a non-invasive liquid biopsy. 

#### 2.4.2. Blood Plasma Markers

##### Proteins

Three main proteins present in blood samples have been used for decades in the diagnosis, prognosis and surveillance of TGCTs: α-fetoprotein (AFP), the β subunit of human chorionic gonadotropin (β-hCG) and Lactate Dehydrogenase (LDH) [19]. When a testicular tumour is suspected, a measurement of these three molecules in the blood of the patient is taken and the changes in those levels are monitored from the very beginning to the end of the treatment to look for possible changes [19]. Nevertheless, neither of them is specific for TGCT diagnosis so they cannot be considered perfect diagnostic tools. Elevated levels of β-hCG can also indicate a diagnosis of other cancers such as prostate, bladder, urethra or kidney cancers [54]. In addition, only 5–40% of seminoma tumours are positive for this marker and it does not provide prognostic information [55,56]. 

Elevated AFP levels are usually linked to NSGCTs [19,57]. In fact, the detection of higher than usual AFP in a cancer previously described as a pure seminoma indicates mixed characteristics overlooked until that point and alters the diagnosis and therapeutic approach accordingly [19]. However, high levels of AFP have also been observed in hepatocellular carcinomas, cirrhotic livers and patients with hepatitis [57].

LDH values are generally used to assess the prognosis of disseminated non-seminomatous tumours before, during and after treatment [19,58]. While roughly half of all advanced testicular cancer patients present higher than average LDH plasma levels, this biomarker is even less specific for testicular cancer than the previous two, so decisions regarding therapeutic approaches are never taken based on it alone [19]. 

##### Nucleic Acids

Studies over the last two decades have sparked an interest over nucleic acid biomarkers in plasma and blood, specifically micro RNA (miRNA) and circulating tumour DNA (ctDNA) [46,59]. ctDNA is genetic material released into the blood stream from tumour cells through necrosis, apoptosis or other biological processes. Recent publications have argued that its analysis would paint a picture of the status of the tumour at a certain point in time and, while other techniques lack analytic value due to tumour heterogeneity, ctDNA could give information about the more aggressive clones [46]. Moreover, it is quite useful to identify patients with TGCTs even if the usual biomarker levels are normal, either by measuring its blood concentration or analysing its methylation patterns [60]. Nevertheless, it requires the use of very sophisticated and sensitive technology and a highly skilled technical workforce, which makes it difficult to use in clinical settings.

Another type of liquid biopsy would be circulating tumour cells (CTCs), which are separated from the primary tumour during metastasis and enter the blood flow [46]. A recent publication detected CTCs in 18% of TGCT patients and linked it to a more aggressive illness [61]. In general, the detection of these could be considered a sign of active metastasis and thus worse prognosis, but its wider application has not been proven yet. 

Recent findings in this field are related to serum microRNA or miRNA. These are small non-coding RNA molecules of around 22 nucleotides that interact with messenger RNA as a kind of regulation system that takes part in numerous regulatory processes, including malignant transformation [62]. Like i(12p), miR-371a-3p appears in elevated concentrations in 90% of TGCTs [63]. This last miRNA molecule is present in blood plasma and its level variations are more representative than AFP and β-hCG variations for TGCT monitoring. Even though it is not a completely specific marker, its representative value increases when combining its analysis with other miRNA molecules [64]. Other miRNA molecules have been identified as possible markers with diagnostic and prognostic value, as well as being useful to monitor the efficacy of therapies in real time. Plasma levels of miR-371 have been proven to be linked to active malignancy in NSGCT patients that have already undergone chemotherapy [65]. To maximise their prognostic or diagnostic value, miRNA markers are often analysed in clusters. 

The serum biomarkers are mainly used in prognosis of TGCTs, but they can be also used for diagnosis. In fact, if these molecules appear in abhorrent concentrations after an orchiectomy, they can indicate previously overlooked metastatic sites [66]. Furthermore, they allow clinicians to get information without the use of invasive techniques, to paint a moving picture of the development of the disease and guide possible treatment changes. 

#### 2.4.3. Recent Discoveries and Future Prospects

Several attempts have been made to find possible additions to the classically used marker proteins, e.g., Placental Alkaline Phosphatase (PLAP), TRA-1-60, Neuro-specific Enolase (NSE), Lectin-reactive AFP and N-glycans, but most of them have not been as useful as initially thought, due to high false-positive rates (TRA-1-60 and NSE), elevated levels in smokers (PLAP) or the inability to actually deliver in clinical settings [46]. 

There has been some progress regarding histological markers, even though these are only useful after orchiectomy. Some examples of that are the chromatin architectural factors of the HMGA family, the transcriptional repressor PATZ, the developmental regulator GPR30 and the cell cycle regulator Aurora B kinase. These have diagnostic value when analysing the histology of the tumours, and GPR30 and Aurora B Kinase are currently being studied in the hope of creating novel therapeutic approaches [67]. 

Recent in vitro studies show that TR4 transcription factor overexpression could be a useful biomarker for case to case prognosis in pure seminoma [68]. This molecule is linked to the epithelial-mesenchymal transition (EMT), which is a known step in metastatic process, but further studies are needed to validate its usefulness in vivo. However, the most promising field of study in TGCT biomarkers is still that of miRNAs, and multiple groups have focused their research on them. 

The field of study of TGCTs has been underexplored due to its less malignant nature. The further study of possible biomarkers with diagnostic and prognostic value would enable clinicians to better assess the benefits of adjuvant therapies against their drawbacks. In fact, offering adjuvant therapies to all stage I testicular cancers would over-treat those who would not develop further disorders (about 70% of them) [46]. That overtreatment increases the risk of second malignancies, toxicity and cardiovascular disease just in the case of cisplatin-based chemotherapy [22]. Better stratification of patients could lead to a lower need for radiation imaging techniques, which in turn increase the risk of further malignancies [46]. Much remains to be studied; considering the connection of TGCTs with faulty spermatogenesis, it could be interesting to learn more about these mechanisms and their molecular basis to search for possible candidate molecules or therapeutic targets.

### 2.5. Spermatogenesis: Key Points to Consider

Before proceeding with the main topic of interest, it is useful to have a brief reminder about the biogenesis of the male gamete, since there are interesting aspects to take into account when assessing the possible molecular targets, both therapeutic and diagnostic, related to testicular cancer. 

Spermatogenesis takes place in the epithelium of seminiferous tubules with the physical support of Sertoli cells and hormonal support of interstitial Leydig cells. Testosterone production by Leydig cells kick-starts the first division of spermatogonial stem cells. The first meiotic division results into two secondary spermatocytes and the second meiotic division into four round spermatids interwoven by their cytosol and cytoskeletons, because of incomplete cytokinesis.

Four round haploid spermatids are transformed in four spermatozoa (the male gamete) in a process called spermiogenesis. It consists of three main events: nucleus compaction, acrosome formation and flagella development. These events are accompanied by a severe structural restructuring, partial cytoplasm removal and mitochondria reorganisation. During this process, the cells move from the base to the lumen of the seminiferous tubule, where the mature spermatids are released. 

During spermiogenesis, the cytoskeleton plays an important role in the structural reorganisation and biogenesis processes of the acrosome, a pouch-like organelle where the hydrolytic enzymes necessary for the fertilisation of eggs are stored. It results from the fusion of pre-acrosomal vesicles migrating from the trans Golgi network in the anterior region of the nucleus [69]. The cytoskeleton forms the acrosome-acroplaxome-manchette, a structure that is necessary to transport the pre-acrosomal vesicles to the proximity of the nuclear envelope [70,71]. It is a key structure essential for the reorganisation of the cytoplasm, the structural changes of the nucleus itself and the elongation of the cell [71]. There are other relevant cytoskeletal structures in spermiogenesis, like the axoneme of the flagellum. All of them are critical to achieve the correct outcome of elongated spermatids; in cases of structural problems, these result in spermatozoid malformations, also known as teratozoospermia [72].

A huge percentage of people suffering from TGCTs had a history of abnormal spermatogenesis [73]. This implies at least a partial relationship between tumorigenesis and anomalies during spermatogenesis. Preliminary data from our research group in teratospermic mice have shown high levels of PCSK6, an indicator of metastasis risk in various types of cancer. Therefore, there is still a lot to know about spermatogenesis, the molecular pathways that contribute to it and its relation to TGCT development. Considering the lack of specific biomarkers for seminoma, this process could be a good starting point to identify new candidate molecules.

## 3. Proproteine Convertases (PCs) as Molecular Targets for Testicular Cancer

The function of numerous proteins is regulated by intricate mechanisms necessary for the correct functioning of the cell. This includes the post-translational modifications of proteins. With the discovery of proinsulin in 1967, it became clear that some proteins of great biological importance undergo different processing stages before becoming active [74] by additions or deletions in the protein sequence, which alter its structure from an inactive (or less active) to an active state. Different types of enzymes partake in these regulatory systems, including the family of the Proprotein Convertases (PC) [75]. 

PCs are a family of serine endoproteases present in mammals, but more closely related to bacterial kexin than to mammalian trypsin, which made their study harder when they were first discovered [76]. The proteins that comprise this family were given independent names as they were discovered, but to avoid confusion, standard nomenclature was proposed. According to this nomenclature, there are nine members of the proprotein convertase family, named by the abbreviation PCSK (Proprotein Convertase Subtilisin/Kexin) followed by a number from 1 to 9: PCSK1, PCSK2, PCSK3, PCSK4, PCSK5, PCSK6, PCSK7, PCSK8 and PCSK9. In addition, some of them are still commonly known by their previous name: PC1, PC2, Furin, PC4, PC5, PACE4, PC7, SKI-1 and NARC-1, respectively. Their function is dependent on calcium presence and pH, so they can only be active in specific spots of the cell, and some PCs, like PCSK6, have additional regulatory mechanisms [77]. 

PCs take part in the processing of many molecules with biological relevance like peptide hormones, viral proteins and membrane receptors [78]. As their function is to regulate the activity of their substrates, irregular PC levels or activity can affect numerous metabolic pathways. Over time, the different proteins of this family have been linked both with the preservation of human homeostasis and diverse pathological states like endocrinopathies, neoplasia, infectious diseases, atherosclerosis and neurodegenerative diseases [79]. For these reasons, Proprotein Convertases quickly became proteins of interest in biomedical research, both as markers and therapeutic targets. 

### 3.1. PCs and Cancer

PCs have been linked to the pathogenesis of neoplastic transformation, cell proliferation, invasion and metastasis [79,80,81]. The exact processes though which tumours adopt a malignant phenotype are still unclear, but the PCs seem to be linked to the activation of transcription factors, membrane receptors and substrates, such as metalloproteases and adhesion molecules, linked to the invasion potential of different tumours [80,82]. Empirical data have shown the relationship between PCs and the carcinogenesis of various types of human cancer and tumour cell lines. An increase in the expression of certain PCs has been observed in these cases, which has been directly linked to the growth and invasion capability of the cells [80,83]. 

Cell proliferation can increase due to the enhanced activation of molecules like TGF-β, PDGF or IGF-1 and their receptors, which are known substrates of the PC family [84,85]. The metabolic environment of cells is very complex, and alterations in the activation of different PC substrates can have numerous effects on the initiation and development of different tumours. For example, in hypoxic conditions of the tumour microenvironment, cell invasion is promoted by transporting furin from the trans Golgi network to the cell surface [86], and the presence of furin in the cell membrane alters the molecular membrane receptors, causing a change in signal transduction. Something similar occurs when PCSK6 expression is enhanced. This membrane PC processes metalloproteases that degrade the extracellular matrix and promote cell migration and, in cancer, metastasis. Moreover, an overactivation of furin, PCSK5 and PCSK7 promotes VEGF expression, a growth factor that is responsible for angiogenesis and lymphogenesis [87]. 

Considering this information, and since PCs are linked to numerous molecular pathways related to tumour development, they are already considered molecules of interest in molecular marker studies in different types of cancer [88]. The overexpression of specific PCs has been directly linked to cancer of the stomach [89], ovary [90], breast [91], colon [92] and nervous system [93]. For most PCs, testicular cancer remains to be explored; for brevity, in this review we will focus on the two PCs linked to various types of cancer: Furin and PCSK6. 

#### 3.1.1. Furin

Furin, also known as PC3 or PCSK3, is the convertase that is most linked to cancer. In recent years, some studies have been carried out specifically surrounding the relationship of furin to inflammation-related cancer types and the immunosurveillance of tumours [81]. It has been confirmed that a T cell-specific furin knockout leads to an increase of the immunosurveillance of triple negative breast tumours [94]. As the membrane receptor PD-1 is one of substrates of furin, its knockout results in lower T-cell apoptosis, augmented infiltration and the enhanced cytotoxicity of CD8^+^ T-cells. This is especially relevant due to the hardships faced in clinics to treat triple negative breast cancer. Similarly, furin has been identified as a pro-oncogenic driver of KRAS and/or BRAF-mediated ERK kinase-pathway activation, which has been linked to the resistance of colorectal cancer to targeted therapies [89]. As a consequence, a knockout of furin leads to increased response to these mentioned therapies. It is interesting to note that KRAS is linked to TGCT tumour development, so furin could be an interesting molecule to study in this regard [53,92]. In fact, two recent bioinformatic analysis have pointed out a significant increase in furin expression in TGCTs compared to healthy tissue, which only reinforces the notion that this could be a fertile ground for further exploration [95,96].

Furthermore, furin expression has been observed to be activated by neoplasia-inducing factors like hypoxia and UV radiation, as well as anti-tumour molecules like IL-12, which promotes T-cell activation and differentiation during inflammation [81]. In conclusion, it is indisputable that furin plays an important role in cancer that should be further explored.

#### 3.1.2. PCSK6

PCSK6, also known as PC6 or PACE4, is a PC expressed in the outer membrane of some cells and is responsible for the processing of molecules of biological and pathological importance, like membrane receptors and metalloproteases. Metalloproteases degrade the extracellular matrix and are involved in the EMT, which, in relation to cancer, leads to metastasis. Therefore, PCSK6 has been studied as a biomarker with which to assess the risk of metastasis or as a therapeutic target in ovarian [97], breast [98] and prostate cancers [99,100], which are also linked to inflammation. 

Considering the shared relation to inflammation, alterations in PCSK6 levels in the case of TGCTs could be a marker as it is in the cancer types mentioned [97,98,99,100]. Our group has observed alterations in its expression in cases of male infertility associated with sperm morphology [33], which may be related to the development of cancer, but we cannot conclude anything else. 

### 3.2. PC in the Testes

Each PC has its own expression pattern within the different tissues in the organism; whereas furin, PCSK5, PCSK6, PCSK7 and PCSK8 are ubiquitous, other PCs are far more specific [79]. PCSK4 is the most limited of all because its expression is specific to germ cells [79]. 

PCSK4 has only been detected in germ-cells and macrophage-like cells in the ovary, specifically in spermatids and the spermatozoids of the epididymis, so it would be reasonable to think that it might be related to the process of gametogenesis [79,101,102,103]. Furthermore, in PCSK4 knockout mice, the spermatozoids have a high probability of undergoing premature capacitation, which leads to hypersensitive membrane receptor involvement in the acrosome reaction. This, in return, leads to a premature acrosome reaction even before getting to the egg, and mice are infertile [102,104]. 

As outlined in Section 2.5, spermatids undergo huge structural and molecular changes during spermiogenesis, and numerous proteins will be expressed that are not involved in any other metabolic pathway. Some of these proteins will need to undergo post-translational modifications to work properly and many will be natural substrates to PCSK4. Validated examples of this are PACAP, ADAM metalloproteinases and some growth factors [75]. 

It is important to note that although PCSK4 is the only germ-cell specific PC, it is not the only PC related to spermatogenesis. Our research group analysed PC expression in testes and saw that furin, PCSK4 and PCSK6 were under expressed in the testes of teratozoospermic GOPC−/− mice when compared with wild type (unpublished data). This could imply a link between PC function and spermatogenesis malformations, which are also related to TGCT development, as stated in Section 2.5. In addition, PCSK6 overexpression has been used as a marker of increased metastasis risk in prostate, breast and ovarian cancer [98]. 

In summary, the analysis of the relationship between PCs and TGCT development could be useful to identify both biomarkers with diagnostic and prognostic value and to find new therapeutic targets for its treatment. 

## 4. Cancer and Inflammation

The molecular environment created by chronic inflammation has been proven to increase the risk of developing certain types of cancer and affect their growth and malignant transformation [105]. Some of the validated substrates of different PCs are involved in these processes, e.g., proteases, cytokines, growth factors and receptors. Therefore, several studies have been carried out that showed a strong link between certain convertases and inflammation-related malignancies [81]. 

Different types of cancer follow different tumorigenic pathways and have different molecular environments. Then overexpression of different PCs is seen as a predictor for more tumour growth and aggressiveness, but their inhibition sometimes helps with cancer treatment or can have the opposite effect [81]. In this review, we are going to highlight the pro-tumoral aspects of PCs. 

Tumour cells interact with their environment to create conditions suitable for their progression and most of the mediators of these interactions are processed by PCs. The result is the creation of the tumour microenvironment, which commonly presents a lower than usual pH, low nutrients and oxygen and a chronic inflammatory state [86]. 

PC substrates also mediate the interaction between tumour cells and the immune system and mark the boundary difference between the immune-vigilance and immune-escape of the tumour [81]. In this context, the suppression of PCs has been shown to lower T cell exhaustion and immune escape, making them interesting targets for immunotherapy [106]. Some pro-inflammatory PC substrates are found in the tumour necrosis factor superfamily: TNF-α, TWEAK, TGF-β, B-cell activating factor, APRIL, several chemokines and hepcidin [81].

Usually, PC involvement in tumour progression and inflammation seems to work through a positive feedback loop. In fact, pro-inflammatory molecules like TGF-β and other growth factors are simultaneously known PC substrates and indirect inducers [81]. This leads to PC over-expression in multiple inflammation-associated cancer types and makes this an interesting field of study to better understand the intricacies of malignancies and their potential treatments [81]. 

### 4.1. TGCTs and Inflammation

To find better ways to stratify TGCT patients according to their risk and prognosis, several studies have been carried out to look for inflammation biomarkers in these tumours. In 2018, a study identified the prognostic value of the programmed death receptor ligand 1 (PD-L1) and the systemic immune inflammation index (SII) in Germ Cell Tumours [107]. In 2019, another study linked the neutrophil to lymphocyte ratio (NLR), lymphocyte to monocyte ratio (LMR) and SII values to TGCT staging [108]. Through those studies, a higher SII was detected in the most aggressive cases, which links inflammatory responses to TGCT, but it is still unclear whether that enhanced inflammation is the cause of the aggressive phenotype or just its manifestation. Furthermore, a study published in 2020 identified both NLR and LMR levels as useful markers at the point of diagnosis to predict future clinical presentation and found a strong relationship between NLR levels and mortality rates during the follow up after operations [109]. 

A further exploration of immune related studies in testicular cancer is the immune privileged status of the testicle. In fact, spermatogenesis is protected from the immune system in mammals by a blood–testis barrier that regulates the exchange of molecules [110]. In TGCTs, a significant increase of immune cell infiltration is observed, but the barrier mentioned earlier is somewhat maintained, as there is still low antigen specific CD8+ T cell activity and very few NK cells [110]. There may even be a relationship between changes in the immune privileged status of the testis and TGCT tumorigenesis. Distinct cytokine profiles have been described in both GCNIS and TGCT, which are not present in healthy and hypofertile testicular tissue, namely B supporting or T-helper cell type 1 driven cytokines [111]. 

Vascularisation also influences carcinogenesis. In functional healthy testes there is an intricate net of veins, but after neoplastic transformation, pro-angiogenesis factors are secreted and that system is disrupted, creating pockets of hypoxia [112]. This is important in embryonic carcinoma, because its cells proliferate better under hypoxia, which in turn creates larger hypoxia pockets and promotes tumour growth, creating a loop [110]. 

The TME of TGCTs is characterised by a high infiltration of CD3+, CD8+ T, T regulatory and B cells. There is also a high influx of pro-inflammatory and anti-inflammatory molecules, some of them known substrates of PCs like TNF-α and CXCL10 [81,110]. Even with so many immune cells infiltrating the tumour, immune escape is made possible by mechanisms like an enhanced PD-L1 expression, WNT/β-catenin pathway (T cell infiltration inhibition), FasL expression (pro-apoptotic signal for tumour infiltrating lymphocytes or TILs) and androgen receptor erasure in Sertoli cells [110]. This interest in the EMT of TGCT has led to an increased search for prognostic and diagnostic biomarkers, as described by Song et al. in their recent bioinformatic analysis [113]. 

### 4.2. Immunotherapeutic Approaches to TGCT Treatment

Most of the efforts to develop an immunotherapeutic approach to TGCTs have focused on PD-L1 checkpoint inhibitors [111]. PD-L1 is a ligand for programmed death receptor 1 (PD-1). T cell activation, tolerance and immune-mediated tissue damage is regulated through its inhibition and in some cases of cancer, PD-L1 is used by tumour cells to suppress tumour immunity. 

Histological studies that analyse the expression of PD-L1 in TGCTs and the healthy testicle have shown contradictory results. Although some studies suggest that PD-L1 is not expressed in testis or in GCNIS [114], another suggests that its expression is constitutive to the testis and is the molecular pathway used to maintain its privileged immune status [115]. Nevertheless, most studies indicate that there is a higher expression of PD-L1 in TGCTs than in healthy tissue [114,115,116,117]. Furthermore, PD-L1 expression levels are useful prognostic factors. Specifically, tumours with low PD-L1 expression respond better to treatment, whereas high expression is linked with poor prognosis. Moreover, when TME immune cells are analysed, low PD-L1 expression in TILs is linked with a worse prognosis [111]. All of this indicates the importance of this checkpoint in TGCTs and PCs, as the membrane receptor PD-1 is a substrate of furin [118]. Several studies have tested this inhibitory checkpoint as a therapeutic target. The data collected from phase II clinical trials are varied and inconsistent, with previous information described by Kalavska et al. in their 2020 review on the topic [111]. In that same paper, they proposed that immune checkpoint inhibitors may not be sufficient to counteract the immune privileged status of the testicle and tumours, and complementary measures might need to be taken to make the therapeutic approach. Moreover, very few patients are eligible for this kind of study and due to the high recovery-rate of TGCTs, only the most aggressive cases enter clinical trials, so the results achieved might not be representative. 

## 5. Conclusions and Future Prospects

The relationship between TGCTs and the immune system remains unexplored, and it could be interesting to study the role of proteins such as PCs in these diseases. The link between PCs and inflammation-related malignancies is unquestionable, and recently there has been an increase in works regarding TGCTs, inflammation and the immune system. It has been confirmed that numerous substrates of the PCs play a significant role in TGCTs, and PCs have been used as targets to improve the immunosurveillance of tumours, so we believe that the application of that knowledge to TGCTs could be invaluable. An example is the relevance of furin driven pathways in the molecular landscape of TGCTs. In fact, one of the genes related to cryptorchidism and, as a consequence, to TGCTs is KRAS, which has been observed to be enhanced by furin in colorectal cancer [53,92].

Furin is also closely related to the TME of TGCT. This PC is involved in processing the membrane receptor PD-1, which is responsible for the immune privileged status of the testis. PD-1 is enhanced in tumours promoting immuno-escape [110], because the interaction between PD-1 and PD-L1 promotes T cell exhaustion and the inhibition of CD+8 cells, and a furin knockout in T cells induces a better response to treatment in triple negative breast cancer [94]. That could be applied to the TGCT tumour microenvironment, to enhance the limited natural antitumoral response of the testis. Even if this approach does not eliminate the tumour, it could be combined with other immunotherapeutic strategies to enhance their impact. Combining this information with recent publications confirming furin overexpression in TGCTs, what began as a hypothetical connection is gaining more and more relevance [95,96].

PCs have been proven to be involved in many diseases, so we think it will be useful to study their involvement in TGCTs. This research could focus on the presence and relevance of furin and other malignancy-related PCs in the therapy-resistant TGCTs, because these PCs are related to PD-1/PD-L1 checkpoint and the immune-privileged status of the testicle. 

There is a need for novel therapeutic approaches that maintain the effectiveness of cisplatin-based chemotherapy but limit the long-term side effects. To date, no research papers have been published with regard to PCs other than PCSK4 in the testis, and only recently have there been references to them in TGCTs [95,96]. We consider that the further study of these fields, with the information we have about the immunological tumour microenvironment and PC substrates, could contribute to the development of much needed new immunotherapies.

## Figures and Tables

**Figure 1 cancers-14-01633-f001:**
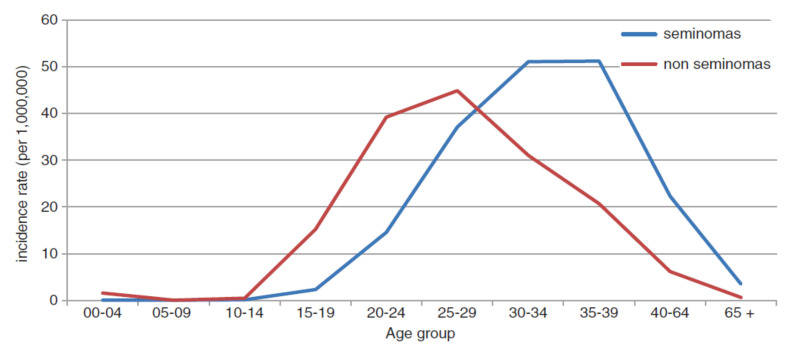
Graph depicting TGCT incidence per 1,000,000 by age group and histology using data from the EUROCARE study, published in 2017 by Annalisa Trama and Franco Berrino [8].

**Table 1 cancers-14-01633-t001:** World Health Organization (WHO) classification of TGCT, adapted [14].

**Germ Cell Tumors Derived from Germ Cell Neoplasia In Situ**
Non-invasive germ cell neoplasia
Germ cell neoplasia in situ (GCNIS)
Specific forms of intratubular germ cell neoplasia
Tumors of single histological type (pure forms)
Seminoma
Seminoma with scyncytiotrophoblast cells
Non-seminomatous germ cell tumors
Embryonal carcinoma
Yolk sac tumor, postpubertal-type
Trophoblastic tumors
Choriocarcinoma
Non-choriocarcinomatous trophoblastic tumors
Placental site trophoblastic tumor
Epithelioid trophoblastic tumor
Cystic trophoblastic tumor
Teratoma, postpubertal-type
Teratoma with somatic-type malignancy
Non-seminomatous germ cell tumors of more than one histologycal type
Mixed germ cell tumors
Germ cell tumors of unknown type
Regressed germ cell tumors
**Germ Cell Tumors Unrelated to GCNIS**
Spermatocytic tumor
Teratoma, prepubertal-type
Dermoid cyst
Epidermoid cyst
Well-differentiated neuroendocrine tumor (monodermal teratoma)
Mixed teratoma and yolk sac tumor, prepubertal-type
Yolk sac tumor, prepubertal-type

**Table 2 cancers-14-01633-t002:** TGCT prognosis classification guidelines by the IGCCCG [43]. The IGCCCG stratifies stage III patients according to three groups: good prognosis, intermediate prognosis, and poor prognosis. AFP: alpha-fetoprotein, hCG: human chorionic gonadotropin, LDH: lactic dehydrogenase and PFS: progression-free survival.

**GOOD PROGNOSIS**	
Non-seminoma	Seminoma
Testis/retroperineal primary tumor And No non-pulmonary visceral metastases And Good markers (all of): AFP< 1000 ng/mL hCG< 5000 iu/L (1000 ng/mL) LDH< 1.5 × upper limit of normal 56% of non-seminomas 5 years PFS 89% 5 year survival 92%	Any primary site And No non-pulmonari visceral metastases And Normal AFP, any hCG and any LDH 90% of Seminomas 5 years PFS 82% 5 year survival 86%
**INTERMEDIATE PROGNOSIS**	
Non-seminoma	Seminoma
Testis/retroperitoneal primary And No non-pulmonary visceral metastases And Intermediate markers (any of): AFP ≥ 1000 and ≤ 10,000 ng/mL hCG ≥ 5000 iu/L and ≤ 50,000 iu/L LDH ≥ 1.5 × N and ≤ 10 × N 28% of non-seminomas 5 years PFS 75% 5 year survival 80%	Any primary siite And Non pulmonary visceral metastases And Normal AFP, any hCG, any LDH 10% of seminomas 5 years PFS 67% 5 year survival 72%
**POOR PROGNOSIS**	
Non-seminoma	Seminoma
Mediastinal primary Or Non-pulmonary visceral metastases Or Poor markers (any of): AFP > 10,000 ng/mL hCG > 50,000 iu/L (10,000 ng/mL) LDH > 10 × upper limit of normal 16% of non-seminomas 5 year PFS 41% 5 year survival 48%	No patients classified as poor prognosis

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
