# Peer review of "Testicular Germ Cell Tumours and Proprotein Convertases"

_cancers, 2022, doi:10.3390/cancers14071633_

Round 1

Reviewer 1 Report

The review by Aitziber Velado-Eguskiza et al. described the involvement of proprotein convertases (PC) in testicular cancer. 

Since the therapy approach in testicular cancer has not changed for years, and it is known that standard therapy has multiple long-term side effects, there is a need for new investigations and conclusions about biomarkers and therapeutic targets. 

Proprotein convertases are already known to be associated with different diseases, cancers, and inflammation, so PCs are well-targeted for this topic of discussion. 

The manuscript is generally well written, with an interesting point of view and worthy-published conclusions. The facts about testicular cancer and the state of the art are well defined in the text. The title is informative and reflects the content, and the abstract is clear and complete. 

Additionally, the authors highlighted the potential relationship between PCs, immune system, inflammation, and testicular cancer, which should be investigated. So, it is clearly shown that PCs should be further investigated as potential biomarkers and/or therapeutic targets.

Author Response

I believe that the reviewer considers the article suitable for publication.

Reviewer 2 Report

The authors reported an up-to-date comprehensive review on the subject of testicular germ cell tumors and proprotein convertases. It is informative to readers. In general, the manuscript is well-organized, however, minor edits could further improve the quality of the manuscript. Overall, the manuscript is acceptable for publication with minor revisions.

Some specific comments and suggestions are as follows:

  1. Previously published reviews on this subject needs to be cited in “Introduction” section of this manuscript, e.g., (1) Batool, A. et al.Testicular germ cell tumor: a comprehensive review.  Mol. Life Sci. 2019, 76: 1713–1727; (2) Mele T, Reid A, Huddart R. Recent advances in testicular germ cell tumours. Fac Rev. 2021, 10:67). Write down a few sentences about the difference, coverage or update from those reviews.
  2. Table 2 should be mentioned in the text and closed to that section.
  3. It is easier for reader to understand if the full names of those abbreviations in Table 2 are given below the table as footnote.
  4. Page 7, line 217, delete “for” between “searched” and “molecules”. Also, what “OCT3/4” stands for?
  5. Page 8, line 261, add “acid’ after “nucleic”.
  6. Page 11, line 432, change “analisis” to “analysis”.
  7. Page 11, section 3.1.2. There is no example (reference) to show PCSK6 is associated with TGCT, although the authors mentioned on the same page, line 417. If the authors believe PCSK6 is a potential biomarker or have unpublished data from their group, it should be discussed here.
  8. The subtitle of section 4 is too broad. Specify the cancer type.
  9. Suggest changing the sub-title of section 5 to “Conclusion and future prospects”.

Author Response

1.- Previously published reviews on this subject needs to be cited in “Introduction” section of this manuscript, e.g., (1) Batool, A. et al.Testicular germ cell tumor: a comprehensive review.  Mol. Life Sci. 2019, 76: 1713–1727; (2) Mele T, Reid A, Huddart R. Recent advances in testicular germ cell tumours. Fac Rev. 2021, 10:67). Write down a few sentences about the difference, coverage or update from those reviews. Interesting. We add the contribution (lines 47-53).

   2.-  Table 2 should be mentioned in the text and closed to that section. Done

  3.-  It is easier for reader to understand if the full names of those abbreviations in Table 2 are given below the table as footnote. That´s alright. We include the abbreviations in the table footnote and also we have improved the table.

 4.- Page 7, line 217, delete “for” between “searched” and “molecules”. done Also, what “OCT3/4” stands for? The transcription factor OCT3/4 (also known as POU5F1 and Oct4) is regarded as one of the key regulators of pluripotency. We add information about this molecule to clarify its meaning (lines 215-218).

 5.- Page 8, line 261, add “acid’ after “nucleic”. Done

  6.- Page 11, line 432, change “analisis” to “analysis”. Done

7.- Page 11, section 3.1.2. There is no example (reference) to show PCSK6 is associated with TGCT, although the authors mentioned on the same page, line 417. If the authors believe PCSK6 is a potential biomarker or have unpublished data from their group, it should be discussed here. Alterations in PCSK6 levels in the case of TGCTs could be a marker as it is in other types of cancers (prostate, breast and ovarian. Referred). Our group has observed alterations in its expression in cases of male infertility associated with sperm morphology, which may be related to the development of cancer (also referenced), but we cannot conclude anything else. An explanation of that is added in the text (lines 444-448).

 8.-   The subtitle of section 4 is too broad. Specify the cancer type. The type of cancer for which the section is being developed is specified below. However, cancer progression and inflammation are processes that occur together in general in many cancers, hence the title of the chapter.

 9.-   Suggest changing the sub-title of section 5 to “Conclusion and future prospects”. It is considered appropriate and changed.

Reviewer 3 Report

This article is an extensive review of testis tumor focused on the rule of PC and inflammation in cancer development , responce to theraphy and prognosis.

Maybe I will reduced the introducing part with the explanation of the histology subtypes, diagnosis blood markers ecc..and focus the review on the second part that is more intersting and novel.

Author Response

Since the aim of this work is to propose proprotein convertases as target molecules either for the detection of testicular germ cell cancer or as therapeutic molecules for its treatment, we consider that it is important to perform an exhaustive analysis of the molecules that to date are related to the progression and stratification of this type of cancer. Likewise, the relevance and the role that the immune system can play may be crucial to understand its progress.